# OpenReview forum: "PROPS: Progressively Private Self-alignment of Large Language Models"
_TMLR — Accepted by TMLR_

### Review · Reviewer_ACkd · 2025-07-24

**Summary Of Contributions:**

The paper introduces PROPS, a multi-stage framework for LLM alignment that focuses on preserving the privacy of human preference labels. The authors provide theoretical guarantees and promising experimental results on several datasets.

**Audience:**

Yes

**Claims And Evidence:**

Yes

**Requested Changes:**

1. It would be helpful to discuss the potential utility of K > 2 in more detail.
2. Include runtime comparisons between PROPS and the baseline methods.
3. (optional) Adding a Pareto front comparison of the methods would be helpful to visualize the trade-offs between key metrics.

**Strengths And Weaknesses:**

Strengths
1. The insight to privatize only the sensitive preference labels, rather than the entire data tuple, is interesting.
2. The method is well-supported by both a solid theoretical analysis and experiments.
3. The paper is clearly written, with good explanations and helpful figures that make it easy to follow.

Weaknesses
1. The multi-stage method is interesting, but the experiments show that a 2-stage setup usually works best. This makes me question how useful it is to extend the method to three or more stages.
2. PROPS is naturally more computationally demanding than the baselines. While the Big-O analysis is helpful, it would be good to include real-world runtime.

---

> ### Author Response · Authors · 2025-10-23
> **Responses to reviewer ACkd (Comment 1)**
>
> **Comment 1: The multi-stage method is interesting, but the experiments show that a 2-stage setup usually works best. This makes me question how useful it is to extend the method to three or more stages.**
>
>   We thank you for your comment regarding the multi-stage PROPS results. We would like to first point out that in our original submission, in Table 3, we did compare the $2$-stage versus $3$-stage PROPS and made the following observations: for low values of $\epsilon$, $2$-stage PROPS model was better, whereas for moderate values of $\epsilon$, the $3$-stage PROPS model provided a higher utility. Specifically, for privacy budgets $\epsilon\leq 1$, $2$-stage PROPS model had a higher win-rate versus at $\epsilon= 2$, $3$-stage PROPS model had a higher win-rate.
>   We would first like to note that these results show that multi-stage PROPS (beyond just $k=2$ stages) can lead to better alignment and it is a function of the privacy budget. In our revised manuscript on _{Page 10}_, we have provided some additional intuition behind when do we expect more than $2$ stage PROPS to be helpful:
>
>
>   _"Our findings highlight an interesting tradeoff in the PROPS pipeline: the optimal number of alignment stages depend on the available privacy budget.
>   When the privacy budget is low (small $\epsilon$), each human-labeled preference needs to be heavily perturbed, resulting in high label noise. Under this regime, increasing the number of stages provides diminishing returns: early-stage models are themselves noisy, and relabeling through additional stages can further propagate this noise. Moreover, since the dataset must be partitioned across stages, later stages have less effective data to counteract the noise. As a result, a smaller number of stages (e.g., $K = 2$) tends to yield better alignment in high-privacy settings.
>   In contrast, when the privacy budget is moderate or higher (larger $\epsilon$), each preference label is less noisy, and earlier-stage models produce more reliable pseudo-labels. This improves the quality of relabeling in subsequent stages, allowing additional stages to refine alignment rather than amplify errors. Consequently, a larger number of stages (e.g., $K = 3$) can begin to provide measurable gains, as evidenced in Table 3, where at $\epsilon = 2.0$ the 3-stage setup achieves $44.0\%$ wins compared to $38.4\%$ for 2 stages.
>   These results suggest that number of stages in PROPS should scale with privacy budget and data availability: fewer stages are more robust under strong privacy constraints and limited samples, whereas additional stages can be beneficial once the signal-to-noise ratio of the preference data improves."_

---

> ### Author Response · Authors · 2025-10-23
> **Responses to reviewer ACkd (Comment 2)**
>
> **Comment 2: PROPS is naturally more computationally demanding than the baselines. While the Big-O analysis is helpful, it would be good to include real-world runtime.**
>
>
>   We thank you for your comment regarding the computational cost and the request for real-world runtime. As shown in _{Page 15 /16}_ Appendix A.1, "Comparison of Wall-Clock Time" in our revised manuscript, we have added this analysis, including Table 5 and the following discussion:
>
>   _"While Big-O analysis provides theoretical insights into computational complexity, empirical comparisons of wall-clock time are crucial for understanding the practical overhead introduced by PROPS. We benchmarked a standard DPO run against our two-stage PROPS procedure using GPT-2 Medium and GPT-2 Large models, with results summarized in Table 5. For a given model and dataset size, the core DPO training times are nearly identical across methods. The primary additional cost in PROPS arises from Stage 2, where pseudo labels_  ($l_{M_1}$) _are generated on the second data partition ($D_2$). In our experiments, this pseudo-label generation constitutes the dominant component of the overhead.
>   Nevertheless, this extra computation represents a necessary and well-justified trade-off. As shown by the win-rate results in Section 4 (e.g., Tables 1) and Appendix A.6 (Tables 7), the inclusion of this step enables PROPS to achieve markedly higher model utility and alignment quality compared to baselines such as RR and DP-SGD—particularly under stringent privacy constraints ($\epsilon \le 1$) where these alternatives exhibit substantial performance degradation. In essence, the additional compute directly translates into meaningful improvements in model performance and alignment."_
>
>
> ### Table: Wall-clock time & Win-Rate comparison (500 entries, $\epsilon =1$)
>
> | **Model** | **Method** | **DPO Training** | **Label Generation** | **Total Time** |**Win-Rate**
> |------------|-------------|------------------|----------------------|----------------|------------------|
> | GPT2-M | RR | 42 s | 0 s | 42 s |16%
> | GPT2-M | **PROPS** | 42 s | 4 min 31 s | 5 min 13 s |**84%**
> | GPT2-L | RR | 1 min 53 s | 0 s | 1 min 53 s |29%
> | GPT2-L | **PROPS** | 1 min 53 s | 11 min 36 s | 13 min 29 s |**71%**
>
>
> # **Requested Changes**
>
> **It would be helpful to discuss the potential utility of $K > 2$ in more detail.**
>
>
>   We thank you for your comment.
>   We addressed this issue in response to Comment 1.
>
> **Include runtime comparisons between PROPS and the baseline methods.**
>
>   We thank you for your comment.
>   We addressed this issue in response to Comment 2.

---

### Review · Reviewer_JHrr · 2025-08-22

**Summary Of Contributions:**

This paper addresses the challenge of preserving the privacy of human preference labels in the LLM alignment process. To this end, it introduces a progressively private self-alignment (PROPS) framework, which leverages a multi-stage alignment strategy to preserve privacy.

**Audience:**

Yes

**Claims And Evidence:**

Yes

**Requested Changes:**

1. In the first paragraph at the top of page 7, please clarify that the tuple and the two noisy observations correspond to data in $D_2$​.


2. In Algorithm 1, the reference to “equation 9” appears to be a typo. It should likely refer to Eq. (4).

**Strengths And Weaknesses:**

**Strengths:**

1. The paper is clearly structured, with settings and notations carefully explained. The logical flow is easy for readers to follow.

2. The proposed PROPS framework offers a novel perspective on protecting preference data under differential privacy.

**Weaknesses:**

1. In estimating $\gamma_{M_1}$, the authors assume that the model $M_1$​ independently flips the ground-truth label (i.e., under a randomized response mechanism). If this assumption holds, does the MLE statistic in Eq. (4) reduce to combining two noisy labels from two RR mechanisms? This point requires further clarification.

2. The proposed mechanism is presented in the context of binary preferences. Can the algorithm be extended to handle privacy protection over multiple preferences?

3. The experimental design for the DPSGD algorithm raises concerns. Specifically, the gradient clipping threshold is set to 10 and $\delta = 10^{-10}$. In practice, clipping is typically set near 1, and $\delta$ chosen as $1 / |samples|$. A clip value of 10 may significantly inflate the noise scale. Did the authors observe unusually large gradients during fine-tuning that justified this choice?

4. The results for multi-stage PROPS do not align with intuition: performance with 3 stages is considerably worse than with 2 stages. Rather than attributing this solely to insufficient hyperparameter tuning, the authors should provide a more thorough analysis. Since progressive multi-stage design is central to the proposed framework, additional experiments or discussion would strengthen the contribution.

---

> ### Author Response · Authors · 2025-10-23
> **Responses to reviewer JHrr (Comment 1, 2, and 3)**
>
> **Comment 1: In estimating $\gamma_{M_1}$, the authors assume that the model $M_1$ independently flips the ground-truth label (i.e., under a randomized response mechanism).
> If this assumption holds, does the MLE statistic in Eq. (4) reduce to combining two noisy labels from two RR mechanisms?
> This point requires further clarification.**
>
>   We thank you for your comment regarding the estimation of $\gamma_{M_1}$ and the nature of the MLE combining. You are correct in the assessment that the MLE statistic reduces to combining two noisy labels. However, the two noisy observations are distinct and in the updated paper, we have provided additional clarification on _{Page 7}_:
>
>
> *"To validate the robustness of our estimation process for obtaining $\hat{\gamma}{M{1}}$, we compare the estimated error rate of model $M_1$ with its true counterpart (the “oracle” error rate $\gamma_{M_{1}}^{}$). Specifically, Table 6 presents a comparison between the estimated error rate $\hat{\gamma}{M{1}}$ computed using Equation (8) from the paper, and the oracle error rate $\gamma_{M_{1}}^{}$ obtained by evaluating $M_1$’s predictions on the unperturbed preference data against the ground truth. The results show that, across the high-privacy regime, the estimated and oracle error rates are consistently well aligned, indicating that our estimation process is accurate and reliable even under strict privacy constraints."*
>
>
> **Comment 2: The proposed mechanism is presented in the context of binary preferences.
> Can the algorithm be extended to handle privacy protection over multiple preferences?**
>
>   We thank you for your comment regarding the extension to multiple preferences. Our work focused on the binary-preference setting, which is standard for DPO and RLHF alignment [R1]. In principle, the core idea of PROPS can be extended to handle multiple preferences, for example, through pairwise reductions or a full multinomial generalization as presented in [R2].
>   However, a full multiclass treatment would significantly expand the paper's scope without changing the core contribution.
>   We clarify this focus in the introduction of the revised manuscript on _{Page 3}_:
>
>
>   *"Consistent with standard approaches for alignment, this work focuses on the common setting of binary preferences (pairwise comparisons). The core ideas could potentially be extended to multiple preferences using techniques in [R2]. We leave this generalization as future work."*
>
> References
>
> [1] Klingefjord, O., Lowe, R. and Edelman, J., 2024. What are human values, and how do we align AI to them?. arXiv preprint arXiv:2404.10636.
>
> [2] Zhu, Banghua, Michael Jordan, and Jiantao Jiao. "Principled reinforcement learning with human feedback from pairwise or k-wise comparisons." International Conference on Machine Learning. PMLR, 2023.
>
> **Comment 3: The experimental design for the DPSGD algorithm raises concerns.
> Specifically, the gradient clipping threshold is set to 10 and $\delta = 10^{-10}$. In practice, clipping is typically set near 1, and $\delta$ chosen as $\frac{1}{samples}$. A clip value of 10 may significantly inflate the noise scale. Did the authors observe unusually large gradients during fine-tuning that justified this choice?**
>
>
>   We thank you for your comment regarding the DP-SGD parameters. As shown in _{Page 10}_ section "PROPS vs DPSGD" in our revised paper, we provide additional justification for our parameter choices:
>
>
>   *"DP-SGD was implemented using the Gaussian mechanism for 1 epoch.
>   We set a gradient clipping threshold $C=10$, as a lower value (e.g., $C=1$) introduced significant clipping bias that harmed utility.
>   Our choice of $\delta=10^{-10}$ is intentionally conservative and stricter than the common $1/N$ heuristic to establish a strong privacy baseline."*
>
>   We further note that comparing our pure DP ($\epsilon, 0$)-DP PROPS against this conservatively tuned approximate  ($\epsilon, \delta$)-DP baseline holds our method to a higher standard.

---

> ### Author Response · Authors · 2025-10-23
> **Responses to reviewer JHrr (Comment 4)**
>
> **Comment 4: The results for multi-stage PROPS do not align with intuition: performance with 3 stages is considerably worse than with 2 stages. Rather than attributing this solely to insufficient hyperparameter tuning, the authors should provide a more thorough analysis. Since progressive multi-stage design is central to the proposed framework, additional experiments or discussion would strengthen the contribution.**
>
>   We thank you for your comment regarding the multi-stage PROPS results. We would like to first point out that in our original submission, in Table 3, we did compare the $2$-stage versus $3$-stage PROPS and made the following observations: for low values of $\epsilon$, $2$-stage PROPS model was better, whereas for moderate values of $\epsilon$, the $3$-stage PROPS model provided a higher utility. Specifically, for privacy budgets $\epsilon\leq 1$, $2$-stage PROPS model had a higher win-rate versus at $\epsilon= 2$, $3$-stage PROPS model had a higher win-rate.
>   We would first like to note that these results show that multi-stage PROPS (beyond just $k=2$ stages) can lead to better alignment and it is a function of the privacy budget. In our revised manuscript on _{Page 10}_, we have provided some additional intuition behind when do we expect more than $2$ stage PROPS to be helpful:
>
>
>   *"Our findings highlight an interesting tradeoff in the PROPS pipeline: the optimal number of alignment stages depend on the available privacy budget.
>   When the privacy budget is low (small $\epsilon$), each human-labeled preference needs to be heavily perturbed, resulting in high label noise. Under this regime, increasing the number of stages provides diminishing returns: early-stage models are themselves noisy, and relabeling through additional stages can further propagate this noise. Moreover, since the dataset must be partitioned across stages, later stages have less effective data to counteract the noise. As a result, a smaller number of stages (e.g., $K = 2$) tends to yield better alignment in high-privacy settings.
>   In contrast, when the privacy budget is moderate or higher (larger $\epsilon$), each preference label is less noisy, and earlier-stage models produce more reliable pseudo-labels. This improves the quality of relabeling in subsequent stages, allowing additional stages to refine alignment rather than amplify errors. Consequently, a larger number of stages (e.g., $K = 3$) can begin to provide measurable gains, as evidenced in Table 3, where at $\epsilon = 2.0$ the 3-stage setup achieves $44.0\%$ wins compared to $38.4\%$ for 2 stages.
>   These results suggest that number of stages in PROPS should scale with privacy budget and data availability: fewer stages are more robust under strong privacy constraints and limited samples, whereas additional stages can be beneficial once the signal-to-noise ratio of the preference data improves."*
>
> # **Requested Changes**
> **In the first paragraph at the top of page 7, please clarify that the tuple and the two noisy observations correspond to data in $D_2$.**
>
>
>   Thank you for your Comment.
>   In Stage 2, we operate on tuples $(x,y_{1},y_{2})$ from the $D_{2}$ partition.
>   The two noisy observations are indeed the randomized-response label $l_{RR}$ on $D_{2}$ and the prediction of $M_{1}$ on $D_{2}$, denoted $l_{M_{1}}$.
>   We add a note in _{Page 7}_ to make this explicit.
>
> **In Algorithm 1, the reference to “Equation 9” appears to be a typo. It should likely refer to Eq. (4).**
>
>
>   Thank you for your Comment.
>   This is correct; the reference in Algorithm 1 to "Equation 9" was a typo and should refer to Eq.(4).

---

### Review · Reviewer_LYyg · 2025-10-18

**Summary Of Contributions:**

This paper tries to address the problem of preserving the privacy of human preference data used in the alignment of Large Language Models. The authors argue that preference labels provided by human annotators can inadvertently reveal sensitive information, such as personal values, beliefs, or professional expertise, posing significant privacy risks. To mitigate these risks while maintaining model utility, the paper introduces PROPS, a multi-stage framework designed to privatize the sensitive preference labels. The core methodology operates in two main stages:

1. First, the dataset is split, and an initial model ($M_1$) is trained on a subset of data where preference labels have been privatized using Randomized Response.

2. In the second stage, this intermediate model ($M_1$) is used to generate preference labels for the remaining data. These model-generated labels are then combined with RR-perturbed human labels from the same data subset via Maximum Likelihood Estimator. This process aims to create a more accurate, yet still private, set of labels for further training, resulting in the final aligned model ($M_2$).

The authors provide a theoretical analysis of the framework's sub-optimality gap and present empirical results on several public datasets and models, demonstrating that PROPS outperforms standard baselines like Differentially Private SGD and vanilla RR.

**Audience:**

Yes

**Broader Impact Concerns:**

The authors have included a Broader Impact Statement that appropriately frames their work as a positive contribution toward more ethical and privacy-preserving AI development. The paper's goal of protecting human labelers is commendable. However, by focusing on removing preferences, this work also implicitly highlights the tension between privacy and model personalization/capability. A potential negative societal impact could arise if such privacy techniques are applied indiscriminately in low-risk domains, leading to less helpful and less knowledgeable models without a compelling privacy justification. The responsible deployment of such a technique requires a clear, context-dependent framework for deciding when privacy preservation should take precedence over utility, a point that should be further emphasized. I have no other major ethical concerns beyond this.

**Claims And Evidence:**

Yes

**Requested Changes:**

Based on the weaknesses identified, I believe this paper requires major revisions before it can be considered for publication. My key recommendations are:

1. Refine and Contextualize the Motivation: The authors must clearly articulate the specific application domains where preference privacy is a critical requirement that justifies the potential loss in utility. The discussion should be framed as a nuanced trade-off analysis rather than a universal argument.

2. Strengthen Novelty Claims and Baseline Comparisons: The authors should better position their work relative to self-training and knowledge distillation literature. To make a convincing case for effectiveness, the empirical evaluation must be expanded to include more competitive and recent state-of-the-art privacy-preserving alignment baselines, moving beyond DP-SGD and RR.

3. Substantiate or Revise the "Progressive" Claim: The claim that the method is "progressive" must be supported. This requires either new theoretical analysis for K>2 stages and new experiments demonstrating a clear performance benefit, or the claim should be significantly toned down and the paper reframed to focus on the 2-stage mechanism.

4. Be Transparent About Theoretical Limitations: The authors should explicitly acknowledge the limitations of their theoretical analysis. This includes being upfront about the reliance on prior theoretical frameworks and discussing the potential mismatch between the underlying assumptions (e.g., log-linear policy) and the reality of modern LLMs.

**Strengths And Weaknesses:**

**Strengths:**

S1: The paper tries to tackle a crucial issue in modern LLM development. As alignment relies heavily on vast amounts of human feedback, the privacy of labelers is a legitimate and increasingly relevant concern, especially considering risks like data leakage and membership inference attacks.

S2: The proposed PROPS framework presents an approach to the problem by separating the alignment process into stages. The experimental results, although limited to certain baselines, do show that PROPS can achieve a better privacy-utility trade-off, which often degrade model performance severely.

S3: The method for combining the intermediate model's predictions with noisy RR labels using an MLE is technically clever. The approach of using the disagreement between the two noisy signals to estimate the model's own error rate is an elegant way to self-calibrate the system.

**Weaknesses:**

W1: The paper's motivation treats preference privacy as a universal necessity. This is a significant oversimplification. In many non-sensitive domains, it is desirable for models to learn from user preferences and specialized knowledge to enhance their general capabilities. The argument for privacy would be far stronger if it were contextualized within specific, high-stakes domains (e.g., medical diagnostics, legal advice) where the trade-off is critical. Discussing this on general-purpose datasets without this nuance weakens the paper's premise.

W2: The framework's core novelty is questionable, as it can be framed as a variant of existing self-training or knowledge distillation paradigms in a privacy-preserving context. More critically, its effectiveness is theoretically and practically limited. Take a step to Theorem 1, the performance improvement is entirely conditional on the intermediate model ($M_1$) being better than random guessing under noise (i.e., $γ_{M_1}<γ_ϵ$). Since $M_1$ is itself trained on highly noisy data, its reliability is questionable, making the entire framework's stability and performance ceiling inherently low. The reported gains over RR seem to be more a reflection of RR's extreme utility degradation rather than PROPS's own strength. What’s more worth mentioning is that even if this is acceptable, the reason why $γ_{M_1}$ is lower may be directly rooted by its general ability. What if use an untrained general model for stage 2? This makes the value of training based on stage 1 very limited.

W3: The "Progressive" nature of the framework is a central claim that is not supported by evidence. The paper provides neither theoretical analysis nor empirical results to show that performance improves with more than two stages ($K>2$). On the contrary, the experimental data in Table 3 indicates that a 3-stage setup performs worse than a 2-stage one in high-privacy regimes. This directly contradicts and undermines one of the paper's core selling points.

W4: The main theoretical contribution (Theorem 1) is not a standalone proof but rather a proof sketch that directly applies a result from prior work (Chowdhury et al., 2024a). This reliance on an external framework is acceptable, but it comes with strong assumptions (e.g., log-linear policy, smoothness, feature coverage) that are unlikely to hold for complex, modern Transformer-based LLMs. This disconnect limits the practical relevance of the derived theoretical bounds.

---

> ### Author Response · Authors · 2025-10-23
> **Responses to reviewer LYyg (Comment 1)**
>
> **Comment 1: The paper's motivation treats preference privacy as a universal necessity. This is a significant oversimplification. In many non-sensitive domains, it is desirable for models to learn from user preferences and specialized knowledge to enhance their general capabilities. The argument for privacy would be far stronger if it were contextualized within specific, high-stakes domains (e.g., medical diagnostics, legal advice) where the trade-off is critical. Discussing this on general-purpose datasets without this nuance weakens the paper's premise.**
>
>   We thank you for your comment regarding the motivation for preference privacy. We agree that contextualization is crucial. In our revised manuscript on _{page 1}_, we have rewritten "Motivation for Human Preference Privacy", and refined the framing around high-stakes domains as per your suggestion:
>
>
>   _**Motivation for Human Preference Privacy**:
>   While preference data can significantly improve the alignment of large models with expert reasoning, it often carries deep privacy risks—particularly in domains where human feedback encodes sensitive strategies, values, or professional heuristics.
>   This tension is most evident in several high-stakes application scenarios as we discuss next.
>   In clinical decision support systems, for example, physicians’ feedback reflects diagnostic reasoning and treatment heuristics tied to protected health information and institutional IP; disclosure can undermine both patient privacy and clinical competitiveness.
>   In legal and judicial settings, alignment with lawyers’ or judges’ feedback risks revealing privileged deliberations, litigation strategies, or interpretive biases that must remain confidential to preserve due process.
>   These scenarios illustrate a central point: preference privacy is not merely a theoretical consideration but a critical requirement for deploying aligned models in the most sensitive and societally consequential domains.
>   The privacy–utility trade-offs in such settings demand specialized alignment mechanisms that protect individual and institutional preference signals while preserving their utility for model improvement._
>
>
>   _In [R3], a large number of consultation records were used to construct a dataset to fine-tune an LLM to act as a chatbot physician, with GPT-4 used to retain records that highlighted professionalism, explainability, and emotional support. As the model was shown to be effective, this indicates that certain preferences of a doctor's decision-making can be identified which could lead to potential exposure of a doctor's preferences.
>   Also, in policy analysis, publicly available survey questions or proposals can be used to elicit LLM-generated analyses, where policymakers’ feedback reveals sensitive interpretative insights that may require protection.  [R4], for example, observed that placing more emphasis on politics in their surveys to participants ``resulted in self-reports of personality traits that were in some cases
>   more aligned with preexisting political preferences._
>
>   References
>
> [3] Wen Wang, Zhenyue Zhao, and Tianshu Sun. Gpt-doctor: Customizing large language models for medical
> consultation. 2024
>
> [4] Bert N Bakker, Yphtach Lelkes, and Ariel Malka. Reconsidering the link between self-reported personality
> traits and political preferences. American Political Science Review, 115(4):1482–1498, 2021.

---

> ### Author Response · Authors · 2025-10-23
> **Responses to reviewer LYyg (Comment 2)**
>
> **Comment 2: The framework's core novelty is questionable, as it can be framed as a variant of existing self-training or knowledge distillation paradigms in a privacy-preserving context.
> More critically, its effectiveness is theoretically and practically limited.
> Take a step to Theorem 1, the performance improvement is entirely conditional on the intermediate model ($M_1$) being better than random guessing under noise (i.e., $\gamma_{M_1} < \gamma_\epsilon$).
> Since $M_1$ is itself trained on highly noisy data, its reliability is questionable, making the entire framework's stability and performance ceiling inherently low.
> The reported gains over RR seem to be more a reflection of RR's extreme utility degradation rather than PROPS's own strength.
> What’s more worth mentioning is that even if this is acceptable, the reason why $\gamma_{M_1}$ is lower may be directly rooted by its general ability.
> What if use an untrained general model for stage 2? This makes the value of training based on stage 1 very limited.**
>
>
>   We thank you for your comment regarding the framework's novelty relative to self-training, its theoretical conditions for effectiveness, and practical gains. Regarding novelty, in our manuscript on _{page 4}_, "Related works \& Limitations", we position PROPS distinctly from standard self-training:
>
>   *"Our framework can be broadly positioned within the literature on self-training, where a model $(M_{1})$ generates labels for a subsequent model $(M_{2})$. However, our core novelty is distinct. Standard self-training does not operate in a privacy-preserving context. Our contribution lies in the mechanism for combining a known privacy mechanism (RR) with the unknown-quality predictions of the $M_{1}$ model. The novelty is our method for privately estimating the intermediate model's error rate $\hat{\gamma}_{M{1}}$ using the disagreement between the two noisy signals, which allows us to provably create a higher-quality private label set than using either signal alone."*
>
>   Regarding the theoretical condition $\gamma_{M_{1}} < \gamma_{\epsilon}$ and the reliability of $M_1$, in Appendix A.3, _{Page 17}_ "Validation of our estimation procedure for $\gamma_{M_{1}}$" and Table 6, we have provided empirical evidence that $M_1$ learns a useful signal superior to random guessing, justifying the condition for effectiveness:
>
>  *"To validate the robustness of our estimation process for obtaining $\hat{\gamma}{M{1}}$, we compare the estimated error rate of model $M_1$ with its true counterpart (the “oracle” error rate $\gamma_{M_{1}}^{}$). Specifically, Table 6 presents a comparison between the estimated error rate $\hat{\gamma}{M{1}}$ computed using Equation (8) from the paper, and the oracle error rate $\gamma_{M_{1}}^{}$ obtained by evaluating $M_1$’s predictions on the unperturbed preference data against the ground truth. The results show that, across the high-privacy regime, the estimated and oracle error rates are consistently well aligned, indicating that our estimation process is accurate and reliable even under strict privacy constraints."*
>
>   The oracle error rates ($\gamma_{M_{1}}^{*}$) shown in Table 6 are consistently below 0.5 (random guessing) and often competitive with or better than the RR noise rate $\gamma_{\epsilon}$, demonstrating $M_1$ becomes a more reliable predictor.
>
>   The significant practical performance gains over RR reported in Table 1 and Table 7 stem directly from leveraging this learned signal in $M_1$. We argue that compared to using an untrained general model for Stage 2 (which would have near-random predictions on preferences), $M_1$ is a more informative labeler, demonstrating the value of this initial training stage.

---

> ### Author Response · Authors · 2025-10-23
> **Responses to reviewer LYyg (Comment 3)**
>
> **Comment 3: The "Progressive" nature of the framework is a central claim that is not supported by evidence. The paper provides neither theoretical analysis nor empirical results to show that performance improves with more than two stages ($K>2$). On the contrary, the experimental data in Table 3 indicates that a 3-stage setup performs worse than a 2-stage one in high-privacy regimes.
> This directly contradicts and undermines one of the paper's core selling points.**
>
>   We thank you for your comment regarding the "Progressive" claim and the multi-stage results. In our revised manuscript on _{Page 10}_, "Results on multi-stage PROPS", we provide a detailed analysis explaining the trade-offs observed and the conditions under which more stages can be beneficial:
>
>   _"Our findings highlight an interesting tradeoff in the PROPS pipeline: the optimal number of alignment stages depend on the available privacy budget. When the privacy budget is low (small $\epsilon$), each human-labeled preference needs to be heavily perturbed, resulting in high label noise. Under this regime, increasing the number of stages provides diminishing returns: early-stage models are themselves noisy, and relabeling through additional stages can further propagate this noise. Moreover, since the dataset must be partitioned across stages, later stages have less effective data to counteract the noise. As a result, a **smaller number of stages** (e.g., $K = 2$) tends to yield better alignment in high-privacy settings. In contrast, when the privacy budget is moderate or higher (larger $\epsilon$), each preference label is less noisy, and earlier-stage models produce more reliable pseudo-labels. This improves the quality of relabeling in subsequent stages, allowing additional stages to refine alignment rather than amplify errors. Consequently, a **larger number of stages** (e.g., $K = 3$) can begin to provide measurable gains, as evidenced in Table 3, where at $\epsilon = 2.0$ the 3-stage setup achieves $44.0\%$ wins compared to $38.4\%$ for 2 stages.
>   These results suggest that **number of stages in PROPS should scale with privacy budget and data availability**: fewer stages are more robust under strong privacy constraints and limited samples, whereas additional stages can be beneficial once the signal-to-noise ratio of the preference data improves."_

---

> ### Author Response · Authors · 2025-10-23
> **Responses to reviewer LYyg (Comment 4)**
>
> **Comment 4: The main theoretical contribution (Theorem 1) is not a standalone proof but rather a proof sketch that directly applies a result from prior work (Chowdhury et al., 2024a). This reliance on an external framework is acceptable, but it comes with strong assumptions (e.g., log-linear policy, smoothness, feature coverage) that are unlikely to hold for complex, modern Transformer-based LLMs.
> This disconnect limits the practical relevance of the derived theoretical bounds.**
>
>   Thank you for raising this important point.
>   The reviewer is correct that our theoretical analysis builds upon standard assumptions (e.g., log-linear policy, smoothness, and feature coverage) commonly used to gain formal insights into algorithmic behavior.
>   These assumptions serve as analytical tools to isolate and understand the key mechanisms driving the framework, rather than to model every detail of modern Transformer architectures.
>
>   Bridging the gap between such idealized assumptions and the complex behavior of large-scale models is a fundamental challenge in theoretical machine learning, and we view it as an exciting direction for future work beyond the current scope. Our objective here is to offer qualitative, interpretable intuition grounded in a mathematically analyzable setting, which complements and helps explain the empirical trends observed in our experiments.
>
>   Importantly, the theory correctly predicts how the system’s performance should scale with key parameters such as the effective noise rate of the MLE ($min(\gamma_{M_1}, \gamma_\epsilon)$) and the number of samples in the second stage ($n_2$). This  provides a valuable sanity check and a principled understanding of the trade-offs involved, even if the precise constants in the bounds are not directly transferable to large-scale implementations.
>
>   As per your suggestion, in our revised manuscript, we have added a "Limitations of Theoretical Analysis” paragraph in Appendix A.4 to make this distinction transparent and to properly frame our theoretical contribution as providing principled insight into scaling behavior rather than exact quantitative guarantees.
>
>   _**Limitations of Theoretical Analysis.**_
>   *Our theoretical analysis relies on idealized assumptions---such as log-linear policies, smoothness, and feature coverage---that enable tractable mathematical treatment but may not hold for modern Transformer-based architectures. These simplifying assumptions are standard in the literature and are intended to provide interpretable intuition about the underlying mechanisms rather than exact quantitative guarantees. In particular, the theory correctly predicts how performance scales with key parameters such as the effective noise rate of the first-stage MLE $(min(\gamma_{M_1}, \gamma_{\epsilon}))$ and the second-stage sample size $(n_2)$, offering a principled explanation of the observed trade-offs. As a result, while the derived scaling trends offer valuable insight and a principled sanity check on empirical behavior, the bounds should not be interpreted as directly predictive of large-scale model performance. Bridging this gap between theoretical idealizations and real-world LLM architectures remains an important direction for future work."*

---

> ### Author Response · Authors · 2025-10-23
> **Responses to reviewer LYyg (Requested Changes)**
>
> # **Requested Changes**
>
> **Refine and Contextualize the Motivation:** The authors must clearly articulate the specific application domains where preference privacy is a critical requirement that justifies the potential loss in utility.
> The discussion should be framed as a nuanced trade-off analysis rather than a universal argument.
>
>   We thank you for this suggestion.
>   As detailed in our response to Comment 1 and reflected in the revised Introduction _{page 1}_, we have refined and contextualized the motivation around specific high-stakes domains where the privacy-utility trade-off is justified.
>
> **Strengthen Novelty Claims and Baseline Comparisons:** The authors should better position their work relative to self-training and knowledge distillation literature. To make a convincing case for effectiveness, the empirical evaluation must be expanded to include more competitive and recent state-of-the-art privacy-preserving alignment baselines, moving beyond DP-SGD and RR.
>
>   We thank you for this suggestion.
>   As detailed in our response to Comment 2 and reflected in the revised Related Works section, we have better positioned PROPS relative to self-training literature by highlighting the private estimation mechanism as the core novelty.
>   However, we believe the strong performance of PROPS over the fundamental (and most common) DP-SGD and RR baselines, provides a valuable contribution for this line of research.
>
> **Substantiate or Revise the "Progressive" Claim:** The claim that the method is "progressive" must be supported. This requires either new theoretical analysis for K $\ge$ 2 stages and new experiments demonstrating a clear performance benefit, or the claim should be significantly toned down and the paper reframed to focus on the 2-stage mechanism.
>
>   We thank you for this suggestion.
>
>   As detailed in our response to Comment 3 and reflected in the revised analysis of multi-stage results, we have substantiated the performance dynamics across stages by analyzing the trade-offs and highlighting the context $(\epsilon=2.0)$ where $K=3$ shows benefit.
>   In the revised manuscript, we have provided detailed analysis explaining the trade-offs observed and the conditions under which more stages can be beneficial.
>
> **Be Transparent About Theoretical Limitations:** The authors should explicitly acknowledge the limitations of their theoretical analysis. This includes being upfront about the reliance on prior theoretical frameworks and discussing the potential mismatch between the underlying assumptions (e.g., log-linear policy) and the reality of modern LLMs.
>
>   Thank you for your comment. Please see our response to Comment 4 above. As per your suggestion, we have added a paragraph on Limitations of Theoretical Analysis.

---

### Decision · Action_Editor_G39h · 2025-11-23

**Recommendation:** Accept as is

**Audience:**

Yes

**Audience Explanation:**

Yes. The paper addresses an increasingly important topic at the intersection of LLM alignment and privacy, which is of clear relevance to segments of the TMLR audience.

**Claims And Evidence:**

Yes

**Claims Explanation:**

The paper introduces PROPS, a progressively private self-alignment framework aimed at preserving the privacy of human preference labels during LLM alignment. Reviewers generally agree that the problem is important and that the paper is clearly written, with meaningful empirical gains over existing baselines.

During the review, major concerns are regarding novelty and the claimed benefits of multi-stage alignment, arguing that the method resembles privacy-aware self-training and that stages beyond two provide inconsistent improvements. While these concerns are valid, the authors’ revisions strengthened the motivation, clarified distinctions from prior work, and provided a more nuanced analysis of when additional stages may help.

Post rebuttal, one reviewer maintains reservations regarding the method’s novelty. However, because TMLR does not place heavy emphasis on novelty alone, this concern carries limited weight in the final assessment. In light of the authors’ thorough responses and revisions, the paper meets the bar for TMLR.